# Full-Gradient Representation for Neural Network Visualization

**Suraj Srinivas**
Idiap Research Institute & EPFL
suraj.srinivas@idiap.ch

**François Fleuret**
Idiap Research Institute & EPFL
francois.fleuret@idiap.ch

## Abstract

We introduce a new tool for interpreting neural net responses, namely full-gradients, which decomposes the neural net response into input sensitivity and per-neuron sensitivity components. This is the first proposed representation which satisfies two key properties: *completeness* and *weak dependence*, which provably cannot be satisfied by any saliency map-based interpretability method. For convolutional nets, we also propose an approximate saliency map representation, called *FullGrad*, obtained by aggregating the full-gradient components.

We experimentally evaluate the usefulness of *FullGrad* in explaining model behaviour with two quantitative tests: pixel perturbation and remove-and-retrain. Our experiments reveal that our method explains model behavior correctly, and more comprehensively, than other methods in the literature. Visual inspection also reveals that our saliency maps are sharper and more tightly confined to object regions than other methods.

## 1 Introduction

This paper studies saliency map representations for the interpretation of neural network functions. Saliency maps assign to each input feature an importance score, which is a measure of the usefulness of that feature for the task performed by the neural network. However, the presence of internal structure among features sometimes makes it difficult to assign a single importance score per feature. For example, input spaces such as that of natural images are compositional in nature. This means that while any single individual pixel in an image may be unimportant on its own, a collection of pixels may be critical if they form an important image region such as an object part.

For example, a bicycle in an image can still be identified if any single pixel is missing, but if the entire collection of pixels corresponding to a key element, such as a wheel or the drive chain, are missing, then it becomes much more difficult. Here the importance of a part cannot be deduced from the individual importance of its constituent pixels, as each such individual pixel is unimportant on its own. An ideal interpretability method would not just provide importance for each pixel, but also capture that of groups of pixels which have an underlying structure.

This tension also reveals itself in the formal study of saliency maps. While there is no single formal definition of saliency, there are several intuitive characteristics that the community has deemed important [1, 2, 3, 4, 5, 6]. One such characteristic is that an input feature must be considered important if changes to that feature greatly affect the neural network output [5, 7]. Another desirable characteristic is that the saliency map must completely explain the neural network output, i.e., the individual feature importance scores must add up to the neural network output [1, 2, 3]. This is done by a redistribution of the numerical output score to individual input features. In this view, a feature is important if it makes a large numerical contribution to the output. Thus we have two distinct notions of feature importance, both of which are intuitive. The first notion of importance assignment is called *local* attribution and second, *global* attribution. It is almost always the case for practical

neural networks that these two notions yield methods that consider entirely different sets of features to be important, which is counter-intuitive.

In this paper we propose full-gradients, a representation which assigns importance scores to both the input features and individual feature detectors (or neurons) in a neural network. Input attribution helps capture importance of individual input pixels, while neuron importances capture importance of groups of pixels, accounting for their structure. In addition, full-gradients achieve this by simultaneously satisfying both notions of *local* and *global* importance. To the best of our knowledge, no previous method in literature has this property.

The overall contributions of our paper are:

1. We show in § 3 that *weak dependence* (see Definition 1), a notion of local importance, and *completeness* (see Definition 2), a notion of global importance, cannot be satisfied simultaneously by any saliency method. This suggests that the counter-intuitive behavior of saliency methods reported in literature [3, 5] is unavoidable.

2. We introduce in § 4 the full-gradients which are more expressive than saliency maps, and satisfy both importance notions simultaneously. We also use this to define approximate saliency maps for convolutional nets, dubbed FullGrad, by leveraging strong geometric priors induced by convolutions.

3. We perform in § 5 quantitative tests on full-gradient saliency maps including pixel perturbation and remove-and-retrain [8], which show that FullGrad outperforms existing competitive methods.

## 2 Related Work

Within the vast literature on interpretability of neural networks, we shall restrict discussion solely to saliency maps or input attribution methods. First attempts at obtaining saliency maps for modern deep networks involved using input-gradients [7] and deconvolution [9]. Guided backprop [10] is another variant obtained by changing the backprop rule for input-gradients to produce cleaner saliency maps. Recent works have also adopted axiomatic approaches to attribution by proposing methods that explicitly satisfy certain intuitive properties. Deep Taylor decomposition [2], DeepLIFT [3], Integrated gradients [1] and DeepSHAP [4] adopt this broad approach. Central to all these approaches is the requirement of *completeness* which requires that the saliency map account for the function output in an exact numerical sense. In particular, Lundberg *et al.*[4] and Ancona *et al.*[11] propose unifying frameworks for several of these saliency methods.

However, some recent work also shows the fragility of some of these methods. These include unintuitive properties such as being insensitive to model randomization [6], partly recovering the input [12] or being insensitive to the model's invariances [5]. One possible reason attributed for the presence of such fragilities is evaluation of attribution methods, which are often solely based on visual inspection. As a result, need for quantitative evaluation methods is urgent. Popular quantitative evaluation methods in literature are based on image perturbation [13, 11, 2]. These tests broadly involve removing the most salient pixels in an image, and checking whether they affect the neural network output. However, removing pixels can cause artifacts to appear in images. To compensate for this, RemOve And Retrain (ROAR) [8] propose a retraining-based procedure. However, this method too has drawbacks as retraining can cause the model to focus on parts of the input it had previously ignored, thus not explaining the original model. Hence we do not yet have completely rigorous methods for saliency map evaluation.

Similar to our paper, some works [3, 14] also make the observation that including biases within attributions can enable gradient-based attributions to satisfy the completeness property. However, they do not propose attribution methods based on this observation like we do in this paper.

## 3 Local *vs.* Global Attribution

In this section, we show that there cannot exist saliency maps that satisfy both notions of *local* and *global* attribution. We do this by drawing attention to a simple fact that $D-$dimensional saliency map cannot summarize even linear models in $\mathbb{R}^D$, as such linear models have $D+1$ parameters. We

prove our results by defining a weak notion of local attribution which we call *weak dependence*, and a weak notion of global attribution, called *completeness*.

Let us consider a neural network function $f : \mathbb{R}^D \to \mathbb{R}$ with inputs $\mathbf{x} \in \mathbb{R}^D$. A saliency map $S(\mathbf{x}) = \sigma(f, \mathbf{x}) \in \mathbb{R}^D$ is a function of the neural network $f$ and an input $\mathbf{x}$. For linear models of the form $f(\mathbf{x}) = \mathbf{w}^T\mathbf{x} + b$, it is common to visualize the weights $\mathbf{w}$. For this case, we observe that the saliency map $S(\mathbf{x}) = \mathbf{w}$ is independent of $\mathbf{x}$. Similarly, piecewise-linear models can be thought of as collections of linear models, with each linear model being defined on a different local neighborhood. For such cases, we can define weak dependence as follows.

**Definition 1.** *(Weak dependence on inputs) Consider a piecewise-linear model*

$$f(\mathbf{x}) = \begin{cases} \mathbf{w}_0^T\mathbf{x} + b_0 & \mathbf{x} \in \mathcal{U}_0 \\ ... \\ \mathbf{w}_n^T\mathbf{x} + b_n & \mathbf{x} \in \mathcal{U}_n \end{cases}$$

*where all $\mathcal{U}_i$ are open connected sets. For this function, the saliency map $S(\mathbf{x}) = \sigma(f, \mathbf{x})$ restricted to a set $\mathcal{U}_i$ is independent of $\mathbf{x}$, and depends only on the parameters $\mathbf{w}_i, b_i$.*

Hence in this case $S(\mathbf{x})$ depends weakly on $\mathbf{x}$ by being dependent only on the neighborhood $\mathcal{U}_i$ in which $\mathbf{x}$ resides. This generalizes the notion of *local* importance to piecewise-linear functions. A stronger form of this property, called *input invariance*, was deemed desirable in previous work [5], which required saliency methods to mirror model sensitivity. Methods which satisfy our weak dependence include input-gradients [7], guided-backprop [10] and deconv [9]. Note that our definition of weak dependence also allows for two disconnected sets having the same linear parameters $(\mathbf{w}_i, b_i)$ to have different saliency maps, and hence in that sense is more general than input invariance [5], which does not allow for this. We now define completeness for a saliency map by generalizing equivalent notions presented in prior work [1, 2, 3].

**Definition 2.** *(Completeness) A saliency map $S(\mathbf{x})$ is*

- *complete if there exists a function $\phi$ such that $\phi(S(\mathbf{x}), \mathbf{x}) = f(\mathbf{x})$ for all $f, \mathbf{x}$.*

- *complete with a baseline $\mathbf{x}_0$ if there exists a function $\phi_c$ such that $\phi_c(S(\mathbf{x}), S_0(\mathbf{x}_0), \mathbf{x}, \mathbf{x}_0) = f(\mathbf{x}) - f(\mathbf{x}_0)$ for all $f, \mathbf{x}, \mathbf{x}_0$, where $S_0(\mathbf{x}_0)$ is the saliency map of $\mathbf{x}_0$.*

The intuition here is that if we expect $S(\mathbf{x})$ to completely encode the computation performed by $f$, then it must be possible to recover $f(\mathbf{x})$ by using the saliency map $S(\mathbf{x})$ and input $\mathbf{x}$. Note that the second definition is more general, and in principle subsumes the first. We are now ready to state our impossibility result.

**Proposition 1.** *For any piecewise-linear function $f$, it is impossible to obtain a saliency map $S$ that satisfies both completeness and weak dependence on inputs, in general.*

The proof is provided in the supplementary material. A natural consequence of this is that methods such as integrated gradients [1], deep Taylor decomposition [2] and DeepLIFT [3] which satisfy completeness do not satisfy weak dependence. For the case of integrated gradients, we provide a simple illustration showing how this can lead to unintuitive attributions. Given a baseline $\mathbf{x}'$, integrated gradients (IG) is given by $\text{IG}_i(\mathbf{x}) = (x_i - x_i') \times \int_{\alpha=0}^{1} \frac{\partial f(\mathbf{x}' + \alpha(\mathbf{x} - \mathbf{x}'))}{\partial x_i} d\alpha$, where $x_i$ is the $i^{th}$ input co-ordinate.

**Example 1.** *(Integrated gradients [1] can be counter-intuitive)*

*Consider the piecewise-linear function for inputs $(x_1, x_2) \in \mathbb{R}^2$.*

$$f(x_1, x_2) = \begin{cases} x_1 + 3x_2 & x_1, x_2 \leq 1 \\ 3x_1 + x_2 & x_1, x_2 > 1 \\ 0 & otherwise \end{cases}$$

*Assume baseline $\mathbf{x}' = (0, 0)$. Consider three points $(2, 2), (4, 4), (1.5, 1.5)$, all of which satisfy $x_1, x_2 > 1$ and thus are subject to the same linear function of $f(x_1, x_2) = 3x_1 + x_2$. However, depending on which point we consider, IG yields different relative importances among the input features. E.g: $IG(x_1 = 4, x_2 = 4) = (10, 6)$ where it seems that $x_1$ is more important (as $10 > 6$),*

*while for $IG(1.5, 1.5) = (2.5, 3.5)$, it seems that $x_2$ is more important. Further, at $IG(2, 2) = (4, 4)$ both co-ordinates are assigned equal importance. However in all three cases, the output is clearly more sensitive to changes to $x_1$ than it is to $x_2$ as they lie on $f(x_1, x_2) = 3x_1 + x_2$, and thus attributions to $(2, 2)$ and $(1.5, 1.5)$ are counter-intuitive.*

Thus it is clear that two intuitive properties of weak dependence and completeness cannot be satisfied simultaneously. Both are intuitive notions for saliency maps and thus satisfying just one makes the saliency map counter-intuitive by not satisfying the other. Similar counter-intuitive phenomena observed in literature may be unavoidable. For example, Shrikumar *et al.* [3] show counter-intuitive behavior of local attribution methods by invoking a property similar global attribution, called *saturation sensitivity*. On the other hand, Kindermans *et al.* [5] show fragility for global attribution methods by appealing to a property similar to local attribution, called *input insensitivity*.

This paradox occurs primarily because saliency maps are too restrictive, as both weights and biases of a linear model cannot be summarized by a saliency map. While exclusion of the bias term in linear models to visualize only the weights seems harmless, the effect of such exclusion compounds rapidly for neural networks which have bias terms for each neuron. Neural network biases cannot be collapsed to a constant scalar term like in linear models, and hence cannot be excluded. In the next section we shall look at full-gradients, which is a more expressive tool than saliency maps, accounts for bias terms and satisfies both weak dependence and completeness.

## 4  Full-Gradient Representation

In this section, we introduce the full-gradient representation, which provides attribution to both inputs and neurons. We proceed by observing the following result for ReLU networks.

**Proposition 2.** *Let $f$ be a ReLU neural network without bias parameters, then $f(\mathbf{x}) = \nabla_{\mathbf{x}} f(\mathbf{x})^T \mathbf{x}$.*

The proof uses the fact that for such nets, $f(k\mathbf{x}) = kf(\mathbf{x})$ for any $k > 0$. This can be extended to ReLU neural networks with bias parameters by incorporating additional inputs for biases, which is a standard trick used for the analysis of linear models. For a ReLU network $f(\cdot; \mathbf{b})$ with bias, let the number of such biases in $f$ be $F$.

**Proposition 3.** *Let $f$ be a ReLU neural network with biases $\mathbf{b} \in \mathbb{R}^F$, then*

$$f(\mathbf{x}; \mathbf{b}) = \nabla_{\mathbf{x}} f(\mathbf{x}; \mathbf{b})^T \mathbf{x} + \nabla_b f(\mathbf{x}; \mathbf{b})^T \mathbf{b} \tag{1}$$

The proof for these statements is provided in the supplementary material. Here biases include both explicit bias parameters and well as implicit biases, such as running averages of batch norm layers. For practical networks, we have observed that these implicit biases are often much larger in magnitude than explicit bias parameters, and hence might be more important.

We can extend this decomposition to non-ReLU networks by considering implicit biases arising due to usage of generic non-linearities. For this, we linearize a non-linearity $y = \sigma(x)$ at a neighborhood around $x$ to obtain $y = \frac{d\sigma(x)}{dx} x + b_\sigma$. Here $b_\sigma$ is the implicit bias that is unaccounted for by the derivative. Note that for ReLU-like non-linearities, $b_\sigma = 0$. As a result, we can trivially extend the representation to arbitrary non-linearities by appending $b_\sigma$ to the vector $\mathbf{b}$ of biases. In general, *any* quantity that is unaccounted for by the input-gradient is an implicit bias, and thus by definition, together they must add up to the function output, like in equation 1.

Equation 1 is an alternate representation of the neural network output in terms of various gradient terms. We shall call $\nabla_{\mathbf{x}} f(\mathbf{x}, \mathbf{b})$ as input-gradients, and $\nabla_b f(\mathbf{x}, \mathbf{b}) \odot \mathbf{b}$ as the bias-gradients. Together, they constitute full-gradients. To the best our knowledge, this is the only other exact representation of neural network outputs, other than the usual feed-forward neural net representation in terms of weights and biases.

For the rest of the paper, we shall henceforth use the shorthand notation $f^b(\mathbf{x})$ for $\nabla_b f(\mathbf{x}, \mathbf{b}) \odot \mathbf{b}$, the bias-gradient, and drop the explicit dependence on $\mathbf{b}$ in $f(\mathbf{x}, \mathbf{b})$.

### 4.1  Properties of Full-Gradients

Here discuss some intuitive properties of full-gradients. We shall assume that full-gradients comprise of the pair $G = (\nabla_x f(\mathbf{x}), f^b(\mathbf{x})) \in \mathbb{R}^{D+F}$. We shall also assume with no loss of generality that

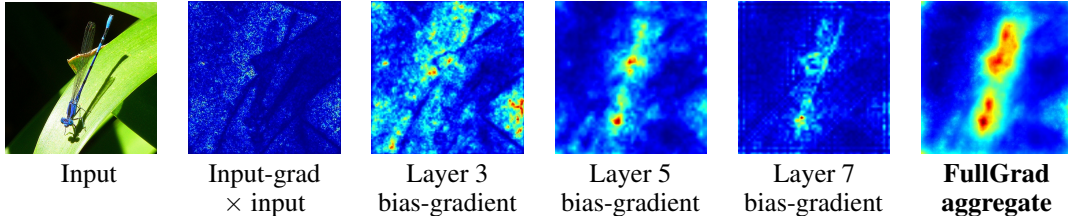

| Input | Input-grad × input | Layer 3 bias-gradient | Layer 5 bias-gradient | Layer 7 bias-gradient | **FullGrad aggregate** |

Figure 1: Visualization of bias-gradients at different layers of a VGG-16 pre-trained neural network. While none of the intermediate layer bias-gradients themselves demarcate the object satisfactorily, the full-gradient map achieves this by aggregating information from the input-gradient and all intermediate bias-gradients. (see Equation 2).

the network contains ReLU non-linearity without batch-norm, and that all biases are due to bias parameters.

**Weak dependence on inputs**: For a piecewise linear function $f$, it is clear that the input-gradient is locally constant in a linear region. It turns out that a similar property holds for $f^b(\mathbf{x})$ as well, and a short proof of this can be found in the supplementary material.

**Completeness**: From equation 1, we see that the full-gradients exactly recover the function output $f(\mathbf{x})$, satisfying completeness.

**Saturation sensitivity**: Broadly, saturation refers to the phenomenon of zero input attribution to regions of zero function gradient. This notion is closely related to global attribution, as it requires saliency methods to look beyond input sensitivity. As an example used in prior work [1], consider $f(x) = a - \text{ReLU}(b - x)$, with $a = b = 1$. At $x = 2$, even though $f(x) = 1$, the attribution to the only input is zero, which is deemed counter-intuitive. Integrated gradients [1] and DeepLIFT [3] consider handling such saturation for saliency maps to be a central issue and introduce the concept of baseline inputs to tackle this. However, one potential issue with this is that the attribution to the input now depends on the choice of baseline for a given function. To avoid this, we here argue that is better to also provide attributions to some function parameters. In the example shown, the function $f(x)$ has two biases $(a, b)$ and the full-gradient method attributes $(1, 0)$ to these biases for input $x = 2$.

**Full Sensitivity to Function Mapping**: Adebayo *et al.* [6] recently proposed sanity check criteria that every saliency map must satisfy. The first of these criteria is that a saliency map must be sensitive to randomization of the model parameters. Random parameters produce incorrect input-output mappings, which must be reflected in the saliency map. The second sanity test is that saliency maps must change if the data used to train the model have their labels randomized. A stronger criterion which generalizes both these criteria is that saliency maps must be sensitive to any change in the function mapping, induced by changing the parameters. This change of parameters can occur by either explicit randomization of parameters or training with different data. It turns out that input-gradient based methods are insensitive to some bias parameters as shown below.

**Example 2.** *(Bias insensitivity of input-gradient methods)*

*Consider a one-hidden layer net of the form $f(x) = w_1 * \text{relu}(w_0 * x + b_0) + b_1$. For this, it is easy to see that input-gradients [7] are insensitive to small changes in $b_0$ and arbitrarily large changes in $b_1$. This applies to all input-gradient methods such as guided backprop [10] and deconv [9]. Thus none of these methods satisfy the model randomization test on $f(x)$ upon randomizing $b_1$.*

On the other hand, full-gradients are sensitive to every parameter that affects the function mapping. In particular, by equation 1 we observe that given full-gradients $G$, we have $\frac{\partial G}{\partial \theta_i} = 0$ for a parameter $\theta_i$, if and only if $\frac{\partial f}{\partial \theta_i} = 0$.

## 4.2   FullGrad: Full-Gradient Saliency Maps for Convolutional Nets

For convolutional networks, bias-gradients have a spatial structure which is convenient to visualize. Consider a single convolutional filter $\mathbf{z} = \mathbf{w} * \mathbf{x} + \mathbf{b}$ where $\mathbf{w} \in \mathbb{R}^{2k+1}$, $\mathbf{b} = [b, b....b] \in R^D$ and $(*)$ for simplicity refers to a convolution with appropriate padding applied so that $\mathbf{w} * \mathbf{x} \in \mathbb{R}^D$, which is often the case with practical convolutional nets. Here the bias parameter is a single scalar $b$ repeated

$D$ times due to the weight sharing nature of convolutions. For this particular filter, the bias-gradient $f^b(\mathbf{x}) = \nabla_{\mathbf{z}} f(\mathbf{x}) \odot \mathbf{b} \in \mathbb{R}^D$ is shaped like the input $\mathbf{x} \in \mathbb{R}^D$, and hence can be visualized like the input. Further, the locally connected nature of convolutions imply that each co-ordinate $f^b(\mathbf{x})_i$ is a function of only $\mathbf{x}[i-k, i+k]$, thus capturing the importance of a group of input co-ordinates centered at $i$. This is easily ensured for practical convolutional networks (e.g.: VGG, ResNet, DenseNet, etc) which are often designed such that feature sizes of immediate layers match and are aligned by appropriate padding.

For such nets we can now visualize per-neuron and per-layer maps using bias-gradients. Per-neuron maps are obtained by visualizing a spatial map $\in \mathbb{R}^D$ for every convolutional filter. Per-layer maps are obtained by aggregating such neuron-wise maps. An example is shown in Figure 1. For images, we visualize these maps after performing standard post-processing steps that ensure good viewing contrast. These post-processing steps are simple re-scaling operations, often supplemented with an absolute value operation to visualize only the magnitude of importance while ignoring the sign. One can also visualize separately the positive and negative parts of the map to avoid ignoring signs. Let such post-processing operations be represented by $\psi(\cdot)$. For maps that are downscaled versions of inputs, such post-processing also includes a resizing operation, often done by standard algorithms such as cubic interpolation.

We can also similarly visualize approximate network-wide saliency maps by aggregating such layer-wise maps. Let $c$ run across channels $c_l$ of a layer $l$ in a neural network, then the FullGrad saliency map $S_f(\mathbf{x})$ is given by

$$S_f(\mathbf{x}) = \psi(\nabla_{\mathbf{x}} f(\mathbf{x}) \odot \mathbf{x}) + \sum_{l \in L} \sum_{c \in c_l} \psi\left(f^b(\mathbf{x})_c\right) \tag{2}$$

Here, $\psi(\cdot)$ is the post-processing operator discussed above. For this paper, we choose $\psi(\cdot) =$ `bilinearUpsample(rescale(abs(·)))`, where `rescale(·)` linearly rescales values to lie between 0 and 1, and `bilinearUpsample(·)` upsamples the gradient maps using bilinear interpolation to have the same spatial size as the image. For a network with both convolutional and fully-connected layers, we can obtain spatial maps for only the convolutional layers and hence the effect of fully-connected layers' bias parameters are not completely accounted for. Note that omitting $\psi(\cdot)$ and performing an additional spatial aggregation in the equation above results in the exact neural net output value according to the full-gradient decomposition. Further discussion on post-processing is presented in Section 6.

We stress here that the FullGrad saliency map described here is approximate, in the sense that the full representation is in fact $G = (\nabla_x f(\mathbf{x}), f^b(\mathbf{x})) \in \mathbb{R}^{D+F}$, and our network-wide saliency map merely attempts to capture information from multiple maps into a single visually coherent one. This saliency map has the disadvantage that all saliency maps have, i.e. they cannot satisfy both completeness and weak dependence at the same time, and changing the aggregation method (such as removing $\odot \mathbf{x}$ in equation 2, or changing $\psi(\cdot)$) can help us satisfy one property or the other. Experimentally we find that aggregating maps as per equation 2 produces the sharpest maps, as it enables neuron-wise maps to vote independently on the importance of each spatial location.

## 5 Experiments

To show the effectiveness of FullGrad, we perform two quantitative experiments. First, we use a pixel perturbation procedure to evaluate saliency maps on the Imagenet 2012 dataset. Second, we use the remove and retrain procedure [8] to evaluate saliency maps on the CIFAR100 dataset.

### 5.1 Pixel perturbation

Popular methods to benchmark saliency algorithms are variations of the following procedure: remove $k$ most salient pixels and check variation in function value. The intuition is that good saliency algorithms identify pixels that are important to classification and hence cause higher function output variation. Benchmarks with this broad strategy are employed in [13, 11]. However, this is not a perfect benchmark because replacing image pixels with black pixels can cause high-frequency edge artifacts to appear which may cause output variation. When we employed this strategy for a VGG-16 network trained on Imagenet, we find that several saliency methods have similar output variation

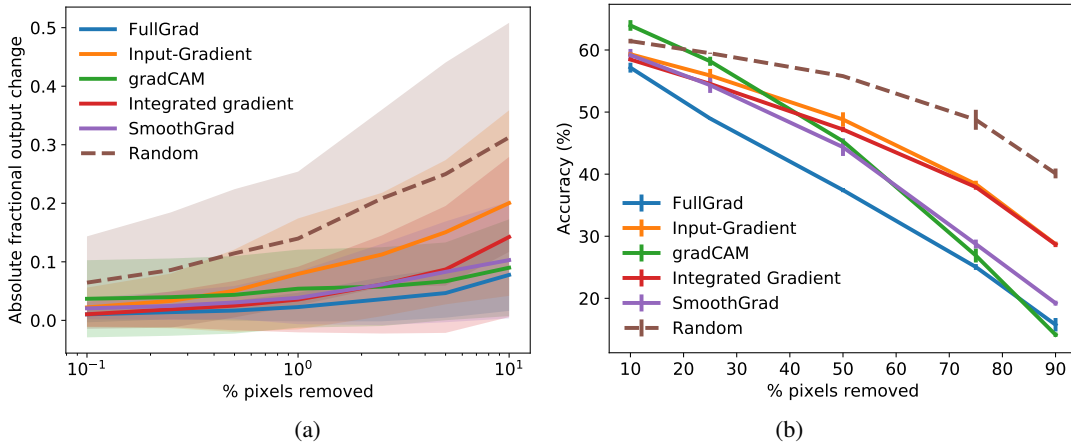

Figure 2: Quantitative results on saliency maps. **(a)** Pixel perturbation benchmark (see Section 5.1) on Imagenet 2012 validation set where we remove $k\%$ least salient pixels and measure absolute value of fractional output change. The **lower** the curve, the better. **(b)** Remove and retrain benchmark (see Section 5.2) on CIFAR100 dataset done by removing $k\%$ most salient pixels, retraining a classifier and measuring accuracy. The **lower** the accuracy, the better. Results are averaged across three runs. Note that the scales of standard deviation are different for both graphs.

to random pixel removal. This effect is also present in large scale experiments by [13, 11]. This occurs because random pixel removal creates a large number of disparate artifacts that easily confuse the model. As a result, it is difficult to distinguish methods which create unnecessary artifacts from those that perform reasonable attributions. To counter this effect, we slightly modify this procedure and propose to remove the $k$ least salient pixels rather than the most salient ones. In this variant, methods that cause the least change in function output better identify unimportant regions in the image. We argue that this benchmark is better as it partially decouples the effects of artifacts from that of removing salient pixels.

Specifically, our procedure is as follows: for a given value of $k$, we replace the $k$ image pixels corresponding to $k$ least saliency values with black pixels. We measure the neural network function output for the most confident class, before and after perturbation, and plot the absolute value of the fractional difference. We use our pixel perturbation test to evaluate full-gradient saliency maps on the Imagenet 2012 validation dataset, using a VGG-16 model with batch normalization. We compare with gradCAM [15], input-gradients [7], smooth-grad [16] and integrated gradients [1]. For this test, we also measure the effect of random pixel removal as a baseline to estimate the effect of artifact creation. We observe that FullGrad causes the least change in output value, and are hence able to better estimate which pixels are unimportant.

## 5.2 Remove and Retrain

RemOve And Retrain (ROAR) [8] is another approximate benchmark to evaluate how well saliency methods explain model behavior. The test is as follows: *remove* the top-$k$ pixels of an image identified by the saliency map for the entire dataset, and *retrain* a classifier on this modified dataset. If a saliency algorithm indeed correctly identifies the most crucial pixels, then the retrained classifier must have a lower accuracy than the original. Thus an ideal saliency algorithm is one that is able to reduce the accuracy the most upon retraining. Retraining compensates for presence of deletion artifacts caused by removing top-$k$ pixels, which could otherwise mislead the model. This is also not a perfect benchmark, as the retrained model now has additional cues such as the positions of missing pixels, and other visible cues which it had previously ignored. In contrast to the pixel perturbation test which places emphasis on identifying unimportant regions, this test rewards methods that correctly identify important pixels in the image.

We use ROAR to evaluate full-gradient saliency maps on the CIFAR100 dataset, using a 9-layer VGG model. We compare with gradCAM [15], input-gradients [7], integrated gradients [1] and a smooth

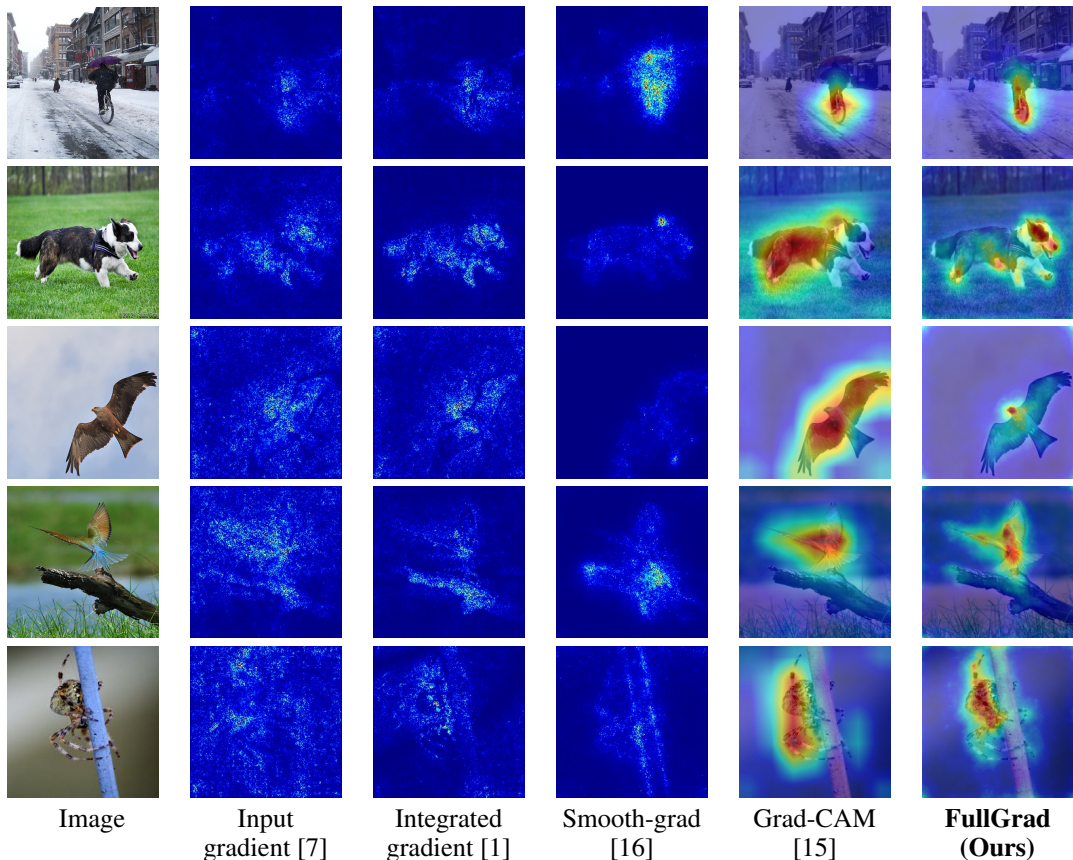

| Image | Input gradient [7] | Integrated gradient [1] | Smooth-grad [16] | Grad-CAM [15] | **FullGrad (Ours)** |
|---|---|---|---|---|---|

Figure 3: Comparison of different neural network saliency methods. Integrated-gradients [1] and smooth-grad [16] produce noisy object boundaries, while grad-CAM [15] indicates important regions without adhering to boundaries. FullGrad combine both desirable attributes by highlighting salient regions while being tightly confined within objects. For more results, please see supplementary material.

grad variant called smooth grad squared [16, 8], which was found to perform among the best on this benchmark. We see that FullGrad is indeed able to decrease the accuracy the most when compared to the alternatives, indicating that they correctly identify important pixels in the image.

### 5.3 Visual Inspection

We perform qualitative visual evaluation for FullGrad, along with four baselines: input-gradients [7], integrated gradients [1], smooth grad [16] and grad-CAM [15]. We see that the first three maps are based on input-gradients alone, and tend to highlight object boundaries more than their interior. Grad-CAM, on the other hand, highlights broad regions of the input without demarcating clear object boundaries. FullGrad combine advantages of both – highlighted regions are confined to object boundaries while highlighting its interior at the same time. This is not surprising as FullGrad includes information both about input-gradients, and also about intermediate-layer gradients like grad-CAM. For input-gradient, integrated gradients and smooth-grad, we do not super-impose the saliency map on the image, as it reduces visual clarity. More comprehensive results without superimposed images for gradCAM and FullGrad are present in the supplementary material.

## 6  How to Choose $\psi(\cdot)$

In this section, we shall discuss the trade-offs that arise with particular choices of the post-processing function $\psi(\cdot)$, which is central to the reduction from full-gradients to FullGrad. Note that by

Proposition 1, any post-processing function cannot satisfy all properties we would like as the resulting representation would still be saliency-based. This implies that any particular choice of post-processing would prioritize satisfying some properties over others.

For example, the post-processing function used in this paper is suited to perform well with the commonly used evaluation metrics of pixel perturbation and ROAR for image data. These metrics emphasize highlighting important regions, and thus the magnitude of saliency seems to be more important than the sign. However there are other metrics where this form of post-processing does not perform well. One example is the digit-flipping experiment [3], where an example task is to turn images of the MNIST digit "8" into those of the digit "3" by removing pixels which provide positive evidence of "8" and negative evidence for "3". This task emphasizes signed saliency maps, and hence the proposed FullGrad post-processing does not work well here. Having said that, we found that a minimal form of post-processing, with $\psi_m(\cdot) = \texttt{bilinearUpsample}(\cdot)$ performed much better on this task. However, this post-processing resulted in a drop in performance on the primary metrics of pixel perturbation and ROAR. Apart from this, we also found that pixel perturbation experiments worked much better on MNIST with $\psi_{mnist}(\cdot) = \texttt{bilinearUpsample}(\texttt{abs}(\cdot))$, which was not the case for Imagenet / CIFAR100. Thus it seems that the post-processing method to use may depend both on the metric and the dataset under consideration. Full details of these experiments are presented in the supplementary material.

We thus provide the following **recommendation to practitioners**: choose the post-processing function based on the evaluation metrics that are most relevant to the application and datasets considered. For most computer vision applications, we believe that the proposed FullGrad post-processing may be sufficient. However, this might not hold for all domains and it might be important to define good evaluation metrics for each case in consultation with domain experts to ascertain the faithfulness of saliency methods to the underlying neural net functions. These issues arise because saliency maps are approximate representations of neural net functionality as shown in Proposition 1, and the numerical quantities in the full-gradient representation (equation 1) could be visualized in alternate ways.

## 7   Conclusions and Future Work

In this paper, we proposed a novel technique dubbed FullGrad to visualize the function mapping learnt by neural networks. This is done by providing attributions to both the inputs and the neurons of intermediate layers. Input attributions code for sensitivity to individual input features, while neuron attributions account for interactions between the input features. Individually, they satisfy *weak dependence*, a weak notion for local attribution. Together, they satisfy *completeness*, a desirable property for global attribution.

The inability of saliency methods to satisfy multiple intuitive properties both in theory and practice, has important implications for interpretability. First, it shows that saliency methods are too limiting and that we may need more expressive schemes that allow satisfying multiple such properties simultaneously. Second, it may be the case that all interpretability methods have such trade-offs, in which case we must specify what these trade-offs are in advance for each such method for the benefit of domain experts. Third, it may also be the case that multiple properties might be mathematically irreconcilable, which implies that interpretability may be achievable only in a narrow and specific sense.

Another point of contention with saliency maps is the lack of unambiguous evaluation metrics. This is tautological; if an unambiguous metric indeed existed, the optimal strategy would involve directly optimizing over that metric rather than use saliency maps. One possible avenue for future work may be to define such clear metrics and build models that are trained to satisfy them, thus being interpretable by design.

## Acknowledgements

We would like to thank Anonymous Reviewer #1 for providing constructive feedback during peer-review that helped highlight the importance of post-processing.

This work was supported by the Swiss National Science Foundation under the ISUL grant FNS-30209.

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
