[Supplementary Material]

# Supplementary material for
# Full-Gradient Representation for
# Neural Network Visualization

## 1 Proof of Incompatibility

**Definition 1.** *(Weak dependence on inputs) Consider a piecewise-linear model*

$$f(\mathbf{x}) = \begin{cases} \mathbf{w}_1^T \mathbf{x} + b_1 & \mathbf{x} \in \mathcal{U}_1 \\ ... \\ \mathbf{w}_n^T \mathbf{x} + b_n & \mathbf{x} \in \mathcal{U}_n \end{cases}$$

*where all $\mathcal{U}_i$ are open connected sets. For this function, the saliency map $S(\mathbf{x}) = \sigma(f, \mathbf{x})$ for some $\mathbf{x} \in \mathcal{U}_i$ is constant for all $\mathbf{x} \in \mathcal{U}_i$ and is a function of only the parameters $\mathbf{w}_i, b_i$.*

**Definition 2.** *(Completeness) A saliency map $S(\mathbf{x})$ is*

- *complete if there exists a function $\phi$ such that $\phi(S(\mathbf{x}), \mathbf{x}) = f(\mathbf{x})$.*

- *complete with a baseline $\mathbf{x}_0$ if there exists a function $\phi_c$ such that $\phi_c(S(\mathbf{x}), S_0(\mathbf{x}_0), \mathbf{x}, \mathbf{x}_0) = f(\mathbf{x}) - f(\mathbf{x}_0)$, where $S_0(\mathbf{x}_0)$ is the saliency of the baseline.*

**Proposition 1.** *For any piecewise-linear function $f$, it is impossible to obtain a saliency map $S$ that satisfies both completeness and weak dependence on inputs, in general*

*Proof.* From the definition of saliency maps, there exists a mapping from $\sigma : (f, \mathbf{x}) \to S$. Let us consider the family of piecewise linear functions which are defined over the same open connected sets given by $\mathcal{U}_i$ for $i \in [1, n]$. Members of this family thus can be completely specified by the set of parameters $\theta = \{\mathbf{w}_i, b_i | i \in [1, n]\} \in \mathbb{R}^{n*(D+1)}$ for $f$ and similarly $\theta'$ for $f'$.

For this family, **weak dependence** implies that the restriction of the mapping $\sigma$ to the set $\mathcal{U}_i$, is denoted by $\sigma|_{\mathcal{U}_i} : (\mathbf{w}_i, b_i) \to S$. Now, since $(\mathbf{w}_i, b_i) \in \mathbb{R}^{D+1}$ and $S \in \mathbb{R}^D$, the mapping $\sigma|_{\mathcal{U}_i}$ is a many-to-one function. This implies that there exists piecewise linear functions $f$ and $f'$ within this family, with parameters $\theta_i = (\mathbf{w}_i, b_i)$ and $\theta'_i = (\mathbf{w}'_i, b'_i)$ respectively (with $\theta_i \neq \theta'_i$), which map to the same saliency map $S$.

**Part (a):** From the first definition of **completeness**, there exists a mapping $\phi : S, \mathbf{x} \to f(\mathbf{x})$. However, for two different piecewise linear functions $f$ and $f'$ that map to the same $S$ for some input $\mathbf{x} \in \mathcal{U}_i$, we must have that $\phi(S, \mathbf{x}) = f(\mathbf{x}) = \mathbf{w}_i^T \mathbf{x} + b_i$ for $f$ and $\phi(S, \mathbf{x}) = f'(\mathbf{x}) = \mathbf{w}'^T_i \mathbf{x} + b'_i$ for $f'$. This can hold for a local neighbourhood around $\mathbf{x}$ if and only if $\mathbf{w}_i = \mathbf{w}'_i$ and $b_i = b'_i$, which we have already assumed to be not true.

**Part (b):** From the second definition of **completeness**, there exists a mapping $\phi_c : S, S_0, \mathbf{x}, \mathbf{x}_0 \to f(\mathbf{x}) - f(\mathbf{x}_0)$.

Let the baseline input $\mathbf{x}_0 \in \mathcal{U}_j$. Similar to the case above, let us assume existence of functions $f$ and $f'$ with parameters $\theta_j = (\mathbf{w}_j, b_j)$ and $\theta'_j = (\mathbf{w}'_j, b'_j)$ respectively (with $\theta_j \neq \theta'_j$), which map to the same saliency map $S_0$. This condition is in addition to the condition already applied on $\mathcal{U}_i$.

Hence we must have that $\phi(S, S_0, \mathbf{x}, \mathbf{x}_0) = f(\mathbf{x}) - f(\mathbf{x}_0) = \mathbf{w}_i^T \mathbf{x} + b_i - \mathbf{w}_j^T \mathbf{x} - b_j$ for $f$ and $\phi(S, S_0, \mathbf{x}, \mathbf{x}_0) = f'(\mathbf{x}) - f'(\mathbf{x}_0) = \mathbf{w}'^T_i \mathbf{x} + b'_i - \mathbf{w}'^T_j \mathbf{x} - b'_j$ for $f'$. This can hold for local

neighbourhoods around $\mathbf{x}$ and $\mathbf{x}_0$ if and only if $\mathbf{w}_i = \mathbf{w}'_i$, $\mathbf{w}_j = \mathbf{w}'_j$ and $b_i - b'_i = b_j - b'_j$. The final condition does not hold in general, hence completeness is not satisfied. $\square$

**When does $b_i - b'_i = b_j - b'_j$ hold?**

- For piecewise linear models without bias terms (e.g.: ReLU neural networks with no biases), the terms $b_i, b'_i, b_j, b'_j$ are all zero, and hence this condition holds for such networks.
- For linear models, (as opposed to piecewise linear models), or when both $\mathbf{x}$ and $\mathbf{x}_0$ lie on the same linear piece, then $b_i = b_j$, which automatically implies that the condition holds.

However these are corner cases and the condition on the biases does not hold in general.

## 2  Full-gradient Proofs

**Proposition 2.** *Let $f$ be a ReLU neural network without bias units, then $f(\mathbf{x}) = \nabla_{\mathbf{x}} f(\mathbf{x})^T \mathbf{x}$.*

*Proof.* For ReLU nets without bias, we have $f(k\mathbf{x}) = kf(\mathbf{x})$ for $k \geq 0$. This is a consequence of the positive homogeneity property of ReLU (i.e; $\max(0, k\mathbf{x}) = k\max(0, \mathbf{x})$)

Now let $\epsilon \in \mathbb{R}^+$ be infinitesimally small. We can now use first-order Taylor series to write the following. $f((1 + \epsilon)\mathbf{x}) = f(\mathbf{x}) + \epsilon f(\mathbf{x}) = f(\mathbf{x}) + \epsilon \mathbf{x}^T \nabla_{\mathbf{x}} f(\mathbf{x})$. $\square$

**Proposition 3.** *Let $f$ be a ReLU neural network with bias-parameters $\mathbf{b} \in \mathbb{R}^F$, then*

$$
\begin{aligned}
f(\mathbf{x}; \mathbf{b}) &= \nabla_{\mathbf{x}} f(\mathbf{x}; \mathbf{b})^T \mathbf{x} + \sum_{i \in [1,F]} (\nabla_b f(\mathbf{x}; \mathbf{b}) \odot \mathbf{b})_i \\
&= \nabla_{\mathbf{x}} f(\mathbf{x}; \mathbf{b})^T \mathbf{x} + \nabla_b f(\mathbf{x}; \mathbf{b})^T \mathbf{b}
\end{aligned}
\tag{1}
$$

*Proof.* We introduce bias inputs $\mathbf{x}_b = \mathbf{1}^F$, an all-ones vector, which are multiplied with bias-parameters $\mathbf{b}$. Now $f(\mathbf{x}, \mathbf{x}_b)$ is a linear function with inputs $(\mathbf{x}, \mathbf{x}_b)$. Proposition applies here.

$$
\begin{aligned}
f(\mathbf{x}, \mathbf{x}_b) &= \nabla_{\mathbf{x}} f(\mathbf{x}, \mathbf{x}_b)^T \mathbf{x} + \nabla_{\mathbf{x}_b} f(\mathbf{x}, \mathbf{x}_b)^T \mathbf{x}_b \\
&= \nabla_{\mathbf{x}} f(\mathbf{x}, \mathbf{x}_b)^T \mathbf{x} + \sum_i (\nabla_{\mathbf{x}_b} f(\mathbf{x}, \mathbf{x}_b))_i
\end{aligned}
\tag{2}
$$

Using chain rule for ReLU networks, we have $\nabla_{\mathbf{x}_b} f(\mathbf{x}, \mathbf{x}_b; \mathbf{b}, \mathbf{z}) = \nabla_z f(\mathbf{x}, \mathbf{x}_b; \mathbf{b}, \mathbf{z}) \odot \mathbf{b}$, where $\mathbf{z} \in R^F$ consists of all intermediate pre-activations. Again invoking chain rule, we have $\nabla_z f(\mathbf{x}, \mathbf{x}_b; \mathbf{b}, \mathbf{z}) = \nabla_b f(\mathbf{x}, \mathbf{x}_b; \mathbf{b}, \mathbf{z})$

$\square$

**Observation.** *For a piecewise linear neural network, $f^b(\mathbf{x})$ is locally constant in each linear region.*

*Proof.* Consider a one-hidden layer ReLU net of the form $f(x) = w_1 * \text{relu}(w_0 * x + b_0) + b_1$, where $f(\mathbf{x}) \in \mathbb{R}$. Let $\rho(z) = \frac{d\text{relu}(z)}{dz}$ be the derivative of the output of relu w.r.t. its inputs. Then the gradients w.r.t. $b_0$ can be written as $\frac{df}{db_0} = w_1 * \rho(w_0 * x + b_0)$. For each linear region, the derivatives of the relu non-linearities w.r.t. their inputs are constant. Thus for a one-hidden layer net, the bias-gradients are constant in each linear region. The same can be recursively applied for deeper networks. $\square$

## 3  Experiments to Illustrate Post-Processing Trade-offs

In this section, we shall describe the experiments performed on the MNIST dataset. First, we perform the digit flipping experiment [1] to test class sensitivity of our method. Next, we perform pixel perturbation experiment as outlined in Section 5.1 of the main paper.

| Method | Random | Gradient | IntegratedGrad | FullGrad | FullGrad (no abs) |
|---|---|---|---|---|---|
| $\Delta$ **log-odds** | $1.41 \pm 8.21$ | $11.92 \pm 17.99$ | $10.81 \pm 20.11$ | $8.26 \pm 21.44$ | $\mathbf{12.93 \pm 18.20}$ |

Table 1: Results on the digit flipping task ($8 \rightarrow 3$). We see that FullGrad (minimal) outperforms others including FullGrad. Larger numbers are better.

| Method | Random | Gradient | IntegratedGrad | FullGrad | FullGrad (no abs) |
|---|---|---|---|---|---|
| **RF = 0.5** | $0.82 \pm 0.28$ | $0.29 \pm 0.22$ | $\mathbf{0 \pm 0}$ | $0.06 \pm 0.13$ | $0.19 \pm 0.19$ |
| **RF = 0.7** | $0.98 \pm 0.34$ | $0.52 \pm 0.27$ | $\mathbf{0.004 \pm 0.06}$ | $0.08 \pm 0.11$ | $0.34 \pm 0.23$ |
| **RF = 0.9** | $1.12 \pm 0.42$ | $0.88 \pm 0.34$ | $\mathbf{0.44 \pm 0.33}$ | $0.55 \pm 0.30$ | $0.63 \pm 0.27$ |

Table 2: Results on the pixel perturbation task on MNIST. In this case, FullGrad performs better than FullGrad (minimal). The overall best performer here is Integrated gradients. Smaller numbers are better.

## 3.1   Digit Flipping

Broadly, the task here is to turn images of the MNIST digit "8" into those of the digit "3" by removing pixels which provide positive evidence of "8" and negative evidence for "3". We perform experiments with a setting similar to the DeepLIFT paper [1], except that we use a VGG-like architecture. Here, FullGrad (no abs) refers to using $\psi_m(\cdot) = \texttt{bilinearUpsample}(\cdot)$ and the FullGrad method refers to using $\psi_m(\cdot) = \texttt{bilinearUpsample(abs}(\cdot))$. From the results in Table 3.1, we see that FullGrad without absolute value performs better in the digit flipping task when compared to FullGrad and all other methods.

## 3.2   Pixel Perturbation

We perform the pixel perturbation task on MNIST. This involves removing the least salient pixels as predicted by a saliency map method and measuring the fractional change in output. The smaller the fractional output change, the better is the saliency method. From Table 3.2, we observe that Integrated gradients perform best overall for this dataset. We hypothesize that the binary nature of MNIST data (i.e.; pixels are either black or white, and "removed" pixels are black) may be well-suited to Integrated gradients, which is not the case for our Imagenet experiments. However, more interestingly, we observe that regular FullGrad outperforms the variant without absolute values.

Thus while for digit flipping it seems that FullGrad (no abs) is the best, followed by gradients and Integrated gradients, for pixel perturbation it seems that Integrated Gradients is the best followed by FullGrad and FullGrad (no abs). Thus it seems that any single saliency or post-processing method is never consistently better than the others, which might point to either the deficiency of the methods themselves, or the complementary nature of the metrics.

# 4   Saliency Results

| Image | Input gradient [2] | Integrated gradient [3] | Smooth-grad [4] | Grad-CAM [5] | **FullGrad (Ours)** |

Figure 1: Comparison of different neural network saliency methods.

| Image | Input gradient [2] | Integrated gradient [3] | Smooth-grad [4] | Grad-CAM [5] | **FullGrad (Ours)** |

Figure 2: Comparison of different neural network saliency methods.