[Reviews · NeurIPS 2019]

Reviewer 1



Updates based on author feedback: Given that the authors added the digit flipping experiments and obtained good results (albeit with different choices of post-processing) I am increasing my score to a 7. However, ***the increased score is based on good faith that the authors will add the following to their paper***: (1) I think it would be extremely helpful to practitioners if the authors exposed how different choices of post-processing affected the results. What happens to the digit flipping experiments when sign information is discarded? What happens to Remove & Retrain when sign information is retained? Please be as up front as possible about the caveats; practitioners should be made aware that the choice of post-processing is something they need to pay close attention to, or the method may be used in a way that gives misleading results (as mentioned, we have seen this happen before with Guided Backprop). I agree that this is a non-trivial problem that should hopefully become a topic of future work. (2) The authors seem to have misunderstood my point regarding gradient*input and LRP. When I said "when bias terms are included in the attributions, the approach of Layerwise Relevance Propagation reduces to gradient*input for ReLU networks and satisfies completeness", I meant that the bias terms are *treated as inputs to the network*. In other words, it is *as though* the "bias" terms are zero (consistent with Proposition 2), because what were formerly bias terms are now included as inputs. The point I was making was that this idea of treating bias terms as inputs to the network, and thus assigning importance to them, has been identified in the literature as the condition under which gradient*input satisfies completeness (note that Layerwise Relevance Propagation satisfies completeness by construction). The authors should therefore cite the prior work where this idea of placing attribution on the bias terms has been discussed. For example, in https://arxiv.org/pdf/1605.01713.pdf they say "We show when all activations are piecewise linear and bias terms are included in the calculation, the Layer-wise Relevance Propagation (LRP) of Bach et al., reduces to gradient*input"; in https://arxiv.org/pdf/1611.07270.pdf they say "In this section, we will show that the z-rule with biases included as input neurons is equivalent to the gradient (Simonyan’s saliency map) multiplied elementwise with the input"; in Figure 2 of https://arxiv.org/pdf/1704.02685.pdf they say "gradient×input assigns a contribution of 10+epsilon_x000F_to x and −10 to the bias term"; thus, the idea of treating biases as inputs to the network and assigning importance to them via gradient*input is not in itself new (the idea of implicit biases is). Original review: ================== Originality I would rate the originality as intermediate-to-high. As mentioned in Section 1, it is known that when bias terms are included in the attributions, the approach of Layerwise Relevance Propagation reduces to gradient*input for ReLU networks and satisfies completeness (https://arxiv.org/abs/1605.01713, https://arxiv.org/abs/1611.07270). However, the idea of extending this to non-ReLU nonlinearities using the concept of "implicit bias" is an interesting creative leap. Having an explanation that is not reduced to some input-level representation is also an interesting idea, although it raises a tradeoff between the complexity of an explanation and how useful it is as an explanation; after all, the activations of all intermediate neurons could be considered an "explanation" for the output of the model, but it's not a very useful explanation. A representation that assigns importance to the bias terms of intermediate neurons is somewhere between a purely input-level representation and merely listing out the activations of all intermediate neurons. I am not sure how useful this "full gradients" representation is, independent of the FullGrad aggregation. Quality I think the quality is overall good, and the ROAR/pixel-perturbation results are quite striking. However, I fear there may be hidden pitfalls of the approach that can only be exposed by more careful benchmarking. The potential pitfalls I am worried about are: (1) As acknowledged in the paper, the FullGrad approach does not provide a way to visualize the influence of fully-connected neurons at the input level. Relatedly, when summing up the bias gradients across several convolutional layers, the FullGrad approach does appear to reflect the full receptive field of the convolutional neurons; instead, it appears that the bias gradients are summed up in an elementwise fashion across different layers. I am concerned that this might result in less precise input-level explanations. (2) The postprocessing operations discard magnitude and sometimes sign information: "we visualize these maps after performing standard post-processing steps that ensure good viewing contrast. These post-processing steps are simple re-scaling operations, often supplemented with an absolute value operation to visualize only the magnitude of importance while ignoring the sign". I am concerned that this means the FullGrad method may not be reliable at precisely identifying which pixels are contributing most positively/negatively to a given task. I would like to see evidence that even after this postprocessing and aggregation, the FullGrad explanations are class-sensitive. Benchmarking using the "digit flipping" experiments from the deepLift/SHAP papers could address this. We have seen from Guided Backprop that aesthetically pleasing visualizations are not necessarily class sensitive; indeed, Guided Backprop performs very poorly on the digit flipping experiments for precisely this reason. (3) The full gradients approach can generate large, discontinuous changes in the explanation over infinitesimal changes in the inputs when switching from one linear regime to another, as with the "thresholding" example in the deepLift paper. To me, these kinds of discontinuous jumps in the explanation are not intuitive, even though they may be necessary to satisfy the "weak dependence" property. That is why I am not completely convinced that "weak dependence" is desirable. A note on the discarding of sign information during post-processing: the authors do say that "One can also visualize separately the positive and negative parts of the map to avoid ignoring signs" - but this raises the question of whether it is safe to aggregate sign information across multiple layers. For example, consider the function y = 10 - z and z = ReLU(b - x), where b is a bias term. It is clear that x contributes positively to y, while the bias term b contributes negatively to y. If bias-gradients assigns a negative importance to b, it might be misleading to aggregate that to an input-level map that may make it appear as though x was contributing negatively. Perhaps that is why the authors suggested taking the absolute value during post-processing, which of course has its own drawbacks. I did not feel that the authors discussed these fairly crucial post-processing choices in enough detail, nor did I feel that I as a reader had much guidance on how to perform this post-processing in my own applications. As a final note on benchmarking, I think that Integrated Gradients should be benchmarked not just with a single reference, but with using multiple sampled references, and the explanations averaged across the sampled references. If runtime becomes an issue, the authors could consider using deepSHAP; averaging of explanations across multiple sampled references is already implemented in the deepSHAP repository and has been found to be beneficial: https://github.com/kundajelab/deeplift#what-should-i-use-as-my-reference. Clarity: I would rate the clarity as good overall. The paper was an enjoyable read. However, I did feel that some crucial details, particularly related to post-processing, were lacking. Also, the text says "the bias-gradient...is shaped like the input...and hence can be visualized like the input...this is easily ensured for practical convolutional networks (e.g.: VGG, ResNet, DenseNet, etc) which are often designed such that feature sizes of immediate layers match and are aligned by appropriate padding" - what about cases where downsampling occurs due to maxpooling or striding? The dimensions of subsequent conv layers would be smaller. Are the bias-gradients just upsampled to make the dimensions match? Or are layers with different dimensions simply not aggregated? I wasn't sure from reading the text. Significance: - The results on ROAR/pixel perturbation are quite impressive, so I would definitely agree that the FullGrad approach has promise for the settings in which it can be applied. But I would like to see more benchmarking before I can confidently recommend the method. I would thus rate the significance as intermediate. My hesitation stems from a concern that the pitfalls I discussed earlier could cause significant problems in practice, especially the garbling of magnitude and/or sign information in some of the post-processing steps; the popularization of Guided Backprop, for example, was quite detrimental in some biology/medicine-related fields because of its tendency to highlight false positives. I would not want the same thing to happen with this method. - I am less certain about the significance of the full gradients representation given the reasons I mentioned earlier, namely that I am not sure how useful such a representation is; the majority of use-cases I can think of seem to require input-level explanations. However, I do think it's a creative way of thinking of the problem.

Reviewer 2



The paper is well-written. The introduction section puts well in perspective the previous work. The difficulty current saliency methods have to achieve both local and global attribution is clearly outlined. The authors propose a simple method that satisfies both global and local properties. It extends gradient x input by viewing the bias as additional input variables that also participate to the gradient x input explanation. Although the biases do not carry any spatial information (they are constant across the feature map), the gradient of the bias does, which implies that the proposed method can be used to further refine the spatial redistribution. The method works surprisingly well empirically, and I think it will be a useful reference method for future developments of explanation techniques. There are however a few minor issues that would be good to address: Batch norm layers can be merged with the adjacent linear/convolution layers. Could the authors verify that the explanation does not change after performing this merging? Definition 2 does not seem to be a sufficient characterization for completeness. Just choose Phi(S(x),x) = f(x), and any saliency map is then complete. In l. 166, there seems to be a "\odot x" missing. The authors should go through the paper to check for potential notation inconsistencies (b vs. x_b, etc.). Example 1 does not seem to be a continuous function. An example involving a continuous function (like deep ReLU networks) would be more convincing. It would be good if the authors mention if their method also satisfy other axioms that have been proposed, e.g. continuity, implementation invariance. I assume it does not. In Figure 1, the bias x gradient at layer 7 seems to be higher resolution than at layer 5. It is counter intuitive as deeper layers typically have less spatial resolution. VGG-16 has many more layer. Why stopping at layer 7? The displayed explanations in Fig. 1 seem to be in absolute value terms. Can the authors confirm that the aggregating has been done on signed heatmaps (which would make most sense) ? Experiments are only on VGG-16. Considering at least one other architecture, e.g. ResNet would be useful.

Reviewer 3



The paper is clearly and well written, with appropriate examples and proofs. It touches an important topic and provides a solution with convincing empirical results. A few concerns/questions: 1. In the pixel perturbation method, the authors propose the idea of removing the last k important features instead of the k most important, motivated by the fact that black pixels can cause high-frequency edge artifacts. I would like to see 1) a better explanation of why the least k features cannot cause high-frequency edge artifacts, and 2) some experiments that make a direct comparison between the 2 versions to back this up empirically. 2. The 2 examples are both showing the downsides of input gradients, while the other methods are not as much illustrated or discussed. For example, Grad-CAM, which is arguably the second best after the proposed FullGrad, is only mentioned in the experiments with no description or discussion. Are these methods suffering from the exact same issues as input gradients or from different issues? 3. The authors mention the sanity check by Adebayo et al. and that in theory FullGrads should fully pass this test, however, does this really happens in practice? For example, they also admit that some signal from the fully connected layers is not fully taken into account (line 236). Would be good to see quantitative results on the existing sanity check. A couple of typos: 102: different --> I think it was meant "same" 155: and well as --> as well as 173: Here "we" discuss

[Author Response · NeurIPS 2019]

We thank all the reviewers for their constructive reviews, and apologize in advance for not being able to answer all
questions or provide detailed experimental results due to the lack of space.

**Digit-flipping experiment (suggested by R1)**: Broadly, the task here is to turn images of the MNIST digit "8" into
those of the digit "3" by removing pixels which provide positive evidence of "8" and negative evidence for "3". We
perform experiments with a setting similar to the DeepLIFT paper, except that we use a VGG-like architecture. The
change in log-odds are as follows for different methods. Random - $1.41 \pm 8.82$ , Gradient - $11.93 \pm 17.87$ , Integrated
gradient - $11.95 \pm 17.19$, FullGrad (with post-processing) - $8.48 \pm 20.81$, FullGrad (raw) - $\mathbf{12.93 \pm 18.21}$. Higher
numbers are better. Here, "raw" FullGrad refers to naive aggregation without using any post-processing, except
up-sampling for max-pooled maps.

We also compare raw FullGrad against the FullGrad with post-processing, on the **pixel perturbation task on MNIST**,
and we find obtain the following fractional output-change values upon removing $70\%$ of least salient pixels - FullGrad
(with post-processing) - $\mathbf{0.08 \pm 0.11}$, FullGrad (raw) - $0.35 \pm 0.23$. Smaller is better here. Similar trend holds for
other fractions of pixel removal and for Imagenet experiments as well. Note that we take absolute value of all heatmaps
as this test requires unsigned heatmaps. Also note that absolute value of raw FullGrad $\neq$ FullGrad with post-processing.

From these experiments, we make the following conclusions: (1) FullGrad explanations are indeed class sensitive (as
measured by digit-flipping), (2) Different interpretability tests may require different different forms of post-processing.
This surprising fact is consistent in spirit with our theory (Proposition 1), which states that **a single saliency map**
**cannot satisfy all intuitive properties we wish to impose**. The digit-flipping experiment emphasizes signed heatmaps,
and hence raw FullGrad does better, while pixel perturbation / ROAR experiments place emphasis on finding important
pixels regardless of their direction of influence, and hence layer-wise post-processing works better here. A principled
method to find suitable post-processing in a task-dependent manner is a non-trivial problem for future work. We
conjecture that this is perhaps analogous to the problem of finding appropriate reference inputs for Integrated gradients /
DeepLIFT (see Q2). Finally, we thank R1 for suggesting this experiment that helped us obtain more insight about these
aspects, and we will add related discussion in an update of the manuscript.

**Response to R1**:

We have responded to concerns about class-sensitivity and the importance of post-processing in the section above.

Q1) *"It is known that when bias terms are included in the attributions, the approach of Layerwise Relevance Propagation*
*reduces to gradient\*input for ReLU networks and satisfies completeness."* While it is true that LRP reduces to gradient \*
input in some cases, **it is incorrect that gradient \* input satisfies completeness for ReLU nets with bias**. Proposition
2 shows that completeness for gradient \* input holds only when the ReLU net has no biases. Completeness with a
baseline also cannot be satisfied as $f(\mathbf{x}) - f(\mathbf{x}_0) = \nabla_{\mathbf{x}} f(\mathbf{x})^T \mathbf{x} + f^b(\mathbf{x}) - \nabla_{\mathbf{x}_0} f(\mathbf{x}_0)^T \mathbf{x}_0 - f^b(\mathbf{x}_0)$, does not reduce
to a gradient \* input term, even when $\mathbf{x}_0 = 0$, as the bias-gradient terms are not equal ($f^b(\mathbf{x}_0) \neq f^b(\mathbf{x})$) in general $\forall \mathbf{x}$
except in certain pathological cases (such as ReLU nets with no bias, or linear models).

Q2) *"Benchmark against Integrated Gradients scores averaged over multiple choices of the reference."* We test this with
pixel perturbation experiments on Imagenet, where we choose references drawn from $N(0,1), N(0,0.1)$ and average
the resulting maps. Specifically, for $1\%$ pixel removal, the fractional change in output is as follows. Zero-reference
$\rightarrow \mathbf{0.035 \pm 0.056}$, $N(0,0.1) \rightarrow 0.042 \pm 0.061$, $N(0,1) \rightarrow 0.065 \pm 0.078$. Smaller is better. Similar trend holds
across other removal fractions. This suggests that we may need better heuristics for reference selection for high
dimensional problems such as Imagenet classification.

**Response to R2**:

Q3) *"Definition 2 does not seem to be a sufficient characterization for completeness."* Thanks for bringing this to our
notice! This can be easily fixed by forcing $\phi$ to depend on $S(\mathbf{x})$, i.e.; by requiring that $\phi(S(\mathbf{x}), \mathbf{x})$ is **not** a constant
function of $S(\mathbf{x})$. This slight modification does not change the proofs or the implications.

**Response to R3**:

Q4) *"I would like to see 1) a better explanation of why the least k features cannot cause high-frequency edge artifacts,*
*and 2) some experiments...to back this up empirically."* (1) Removing least-k features still causes artefacts, but the
test measures which saliency methods identify pixels whose change doesn't affect the output, even if they have
large artefacts. For the largest-k feature removal test, **random heatmaps would create maximal artefacts without**
**identifying important pixels**, and can cause large output changes and falsely be considered a strong baseline. (2) For
largest-k test, with $1\%$ pixel removal, we obtain following output-change values. Random - $0.14 \pm 0.11$, Integrated
gradients - $0.15 \pm 0.11$. Here larger is better, and it (falsely) seems that both methods are almost identical. For least-k,
random remains the same, but Integrated gradient gets $0.036 \pm 0.056$. Here smaller is better, and the gap between the
methods is more evident. We will add related discussion in an update of the manuscript.

[Meta-Review · NeurIPS 2019]

All the reviewers agree the paper should be accepted. The rebuttal added significant information and experiments, which must be added to the camera ready copy. The paper is accepted on this condition.